# Children Growing Up with Severe Disabilities as a Result of Snakebite Envenomations in Indigenous Villages of the Brazilian Amazon: Three Cases and Narratives

**DOI:** 10.3390/toxins15060352

**Published:** 2023-05-23

**Authors:** Altair Seabra de Farias, Joseir Saturnino Cristino, Macio da Costa Arévalo, Alceonir Carneiro Junior, Manoel Rodrigues Gomes Filho, Sediel Andrade Ambrosio, João Nickenig Vissoci, Fan Hui Wen, Vinícius Azevedo Machado, Jacqueline Sachett, Wuelton Monteiro

**Affiliations:** 1Escola Superior de Ciências da Saúde, Universidade do Estado do Amazonas, Manaus 69065-001, Brazil; asfarias@uea.edu.br (A.S.d.F.); joseysaturnino@gmail.com (J.S.C.); vmachado@uea.edu.br (V.A.M.); jac.sachett@gmail.com (J.S.); 2Diretoria de Ensino e Pesquisa, Fundação de Medicina Tropical Dr. Heitor Vieira Dourado, Manaus 69040-000, Brazil; 3Distrito Sanitário Especial Indígena Alto Rio Solimões, Secretaria Especial de Saúde Indígena, Tabatinga 69640-000, Brazil; macio.arevalo@saude.gov.br (M.d.C.A.); enfermeiromanoel01@gmail.com (M.R.G.F.); 4Distrito Sanitário Especial Indígena Manaus, Secretaria Especial de Saúde Indígena, Manaus 69050-010, Brazil; altair17@usp.br; 5Faculdade de Medicina, Universidade Federal do Amazonas, Manaus 69020-160, Brazil; sedyell@hotmail.com; 6Department of Emergency Medicine, Duke University School of Medicine, Durham, NC 27710, USA; jnv4@duke.edu; 7Duke Global Health Institute, Duke University, Durham, NC 27708, USA; 8Instituto Butantan, São Paulo 05503-900, Brazil; fan.hui@butantan.gov.br; 9Diretoria de Ensino e Pesquisa, Fundação Alfredo da Matta, Manaus 69065-130, Brazil

**Keywords:** indigenous peoples, access to health care, Brazilian Amazon, disabilities

## Abstract

Snakebites are a major public health problem in the Brazilian Amazon and may lead to local complications and physical deficiencies. Access to antivenom treatment is poorer in indigenous populations compared to other populations. In this study, we report three cases of long-term severe disabilities as a result of *Bothrops atrox* snakebites in indigenous children, according to the narratives of the parents. The three cases evolved to compartment syndrome, secondary bacterial infection and extensive necrosis. The cases are associated with delayed antivenom treatment due to very fragmented therapeutic itineraries, which are marked by several changes in means of transport along the route. The loss of autonomy at such an early stage of life due to a disability caused by a snakebite, as observed in this study, may deprive children of sensory and social experiences and of learning their future roles in the community. In common to all cases, there was precarious access to rehabilitation services, which are generally centralized in the state capital, and which leads to a prolonged hospitalization of patients with severe snakebite, and distances them from their territory and family and community ties. Prospective studies should be conducted in the Amazon that estimate the burden of disabilities from snakebites in order to formulate public policies for the treatment and rehabilitation of patients through culturally tailored interventions.

## 1. Introduction

All children have the right to survive and develop and to live a life free from disease, illnesses or other conditions that affect their well-being and future prospects [1]. However, there are nearly 240 million children with disabilities in the world, though many of these are preventable through adequate nutrition, immunization or adequate access to the health system [2]. Children living in low- and middle-income countries (LMIC) are more likely to be affected by these preventable conditions that can lead to disabilities [2]. Families of children with disabilities may experience stigma and tend to perform health care for their children beyond that performed by other families [3,4,5]. The extent of these additional concerns varies among families and depends on the nature and severity of the children’s disabilities and the social context in which the meaning of the children’s disabilities is interpreted [3]. 

Snakebite envenomation is an important cause of long-term disabilities in LMIC countries [6,7,8]. Cases of disabilities in children that are caused by snakebites are reported all around the world, with higher significant physical, psychological and economic impacts than in adults, given their longer life expectancy [9,10,11]. In Brazil, approximately 30,000 cases of snakebites are registered per year, with 15% occurring in children under 14 years of age [12]. Snakebite incidence is five-fold higher in the Brazilian Amazon compared to the rest of the country [11]. Furthermore, in the Amazon region of Brazil, children are proportionally more affected than in the other regions of the country [13]. In Brazil, living in rural areas and time to care of >3 h are risk factors for severity following snakebites in children [14]. This epidemiological context is precisely what we find in the Amazon, where cultural and geographical factors contribute to poor access to hospitals for antivenom treatment, especially riverine and indigenous populations [15,16,17,18]. Regardless of the noteworthy higher incidence of snakebites in children in the Brazilian Amazon, the number of intensive care units and pediatricians is proportionately lower in this region [14]. 

In the Amazon, *Bothrops atrox* is the major cause of snakebite envenomations [19]. Concerning local effects, *B. atrox* envenomations lead to tissue damage caused by intravascular coagulation, rupture of blood capillary vessels and digestion of the extracellular matrix [20,21,22,23,24]. These effects manifest themselves clinically in signs and symptoms such as pain, swelling, regional lymphadenopathy, ecchymosis, blistering, and necrosis [19]. Severe cases can progress to compartment syndrome, atrophy, loss of limb function and even amputation [25,26,27].

A national survey in Brazil demonstrated that the indigenous population has a significantly higher prevalence of motor disabilities in relation to other groups [28]. Similarly, other studies demonstrate a much higher rate of snakebite envenomations in indigenous populations of the Brazilian Amazon [11,29]. This ecological association indicates the urgent need to investigate the burden of disabilities caused by snakebites among indigenous populations. In general, previous studies carried out in the Amazon on disabilities caused by snakebites have described only the clinical aspects of the problem [19,25,26,27], without stressing the repercussions of the problem on the social life of patients. In this study, which seeks to address this problem, we report the clinical characteristics and narratives obtained from interviews with the parents and with three children presenting severe physical disabilities from snakebite envenomations in indigenous villages of the Brazilian Amazon, which were identified in previous surveys carried out in these areas [13]. 

## 2. Case Reports

The characteristics of the children with disabilities and therapeutic itinerary are presented in Table 1. 

In Figure 1, it is possible to observe the trajectory taken by each child who was the victim of a snakebite from the moment of the bite until they reach specialized care.

### 2.1. Case 1

A 10-year-old boy of Apurinã ethnicity, resident of the Vila Nova community in the municipality of Tapauá, Amazonas, was bitten by a jararaca (*Bothrops atrox*) when he was 9 years old. The interview we conducted was with the patient’s mother, who accompanied her son to Manaus for treatment; she is 28 years old and speaks Portuguese. When the questions were addressed to her son, she served as an interpreter, since he is only fluent in Apurinã, despite the fact that the school in which he studies is bilingual (Portuguese-Apurinã). The mother is not married, and lives with her two children; the boy and a girl who is a year younger, and with her parents and her brothers and sisters. The boy’s father, however, also lives in the village. The family lives off fishing and farming. According to the interviewee, the main fish they catch are peacock bass, sardines, pacu, arowana and tambaqui. The main crops are fruits such as banana and pineapple, cassava and cará (*Dioscorea trifida*). The village is located eight hours from the city of Tapauá by river using a speedboat. According to the interviewee, the houses are located very close to the forest.

The boy was bitten while harvesting pineapples in the plantation together with his mother, sister and aunt. At the time of the bite, he was alone, at a certain distance from the rest of the family. According to the boy’s reports, translated by his mother, he walked past a fallen log and stepped on the snake, which then bit him. The snake was wrapped around the log, and it was a large, dark snake, the boy said. After receiving this first bite, he felt himself being bitten on the other foot as well. According to him, at this time, there were several bites because “the large snake was accompanied by several hatchlings”. After these bites, the boy saw what was causing the bites: a large snake accompanied by several hatchlings. Then, he started running and screaming for his mother. He ran until he reached the place where the mother was working and reported what had happened; even with pain and fear he shouted: “It’s a snake, mother!”. The mother then went out to look for the snakes, but could not find them. From the mother’s account, they were very diligent in searching for the snake, but as they did not find it, she concluded that it had “gone into a hole in the ground”. When asked which snake most likely bit the boy, she immediately answered: “We call it a jararaca”. She adds that jararaca bites are common in the village, even in children: “It happened to a child of my brother, also 5 years old. He also came to Manaus by speedboat. He was bitten in the forest, on a trail. He was bitten three times in the foot”. However, in this case, she points out that the snake was seen by the adult accompanying the child, and that the snake was killed. “But I didn’t kill this one”, she said, referring to the snake that bit her son.

After concluding that the perpetrating snake would not be found, the mother said she examined the boy’s lower limbs and saw several spots of snake fang marks; more precisely, she said the boy “was bitten eight times, on both legs”. No treatment was given to the boy during this time and soon they returned to the village, on a route that lasted approximately 1 h by canoe. They arrived in the village at approximately 5 pm. In the village, the mother asked for help from the tribal chief, who is the political leader of the community. The only medicine used was some animal lard, which the mother did not know from what animal it was. Afterwards, the boy himself cleaned the bites just with water, removing dirt and blood. According to the mother, the boy was not given anything to drink or eat: “You mustn’t!”. When the mother was asked about what the boy felt, she translates the question to the boy. Translating the boy’s response, she replies: “He felt pain, right?” According to her, “He didn’t talk anymore, he stopped talking. His whole legs were swollen; the whole body was swollen”. The boy’s mother reports that the villagers do not like to go to the city to take care of their health problems; they always prefer to stay in the village. In this case, the son had to be taken to the city because he felt a lot of pain.

The chief contacted the indigenous health base via radio. The boy’s transport, however, only occurred at 3 pm the next day, and he arrived at the indigenous health unit at 7 pm. The person who took care of the boy during the journey was the chief. This unit does not have antivenom available. An indigenous health worker gave the boy a painkiller. According to the mother, it was a medicine “for pain and to make the tiredness go away”. An uncle accompanied the boy to the hospital. In a serious condition, the boy was transferred by ambulance to the hospital of Tapauá, where he received the anti-*Bothrops* serum approximately 30 h after the bite. At the hospital, the severity of the boy’s condition was verified, and transfer to Manaus by plane was recommended. Via radio, the mother was warned in the village about the transfer, and began her journey to Manaus to accompany her son’s treatment.

In Manaus, the boy underwent surgical treatment and, due to extensive necrosis, suffered great loss of tissue of the right lower limb, and it was necessary to amputate the 3rd, 4th and 5th right pododactyls. He also had a secondary bacterial infection and had compartment syndrome and required fasciotomy and debridement. He remained in Manaus for six months for wound care, with several surgical debridement procedures and larval therapy aimed at better healing, as well as physiotherapy aimed at restoring part of the mobility of the limb (Figure 2). 

### 2.2. Case 2

A 9-year-old boy of Tikuna ethnicity, living in the Novo Cruzador community in the municipality of Tabatinga, Amazonas, was bitten by a jararaca when he was 7 years old. The interview conducted for this study was conducted with the patient’s father, with the help of a Tikuna interpreter. The boy was bitten while he went alone to defecate on the edge of the creek, in an area of forest. The father reports that the child screamed and relatives went to help him. A neighbor also went to the scene and killed the snake, which was identified as a pregnant jararaca. During first aid in the village, salt was used with water at the site of the bite to make the pain go away. The bite occurred at approximately 9 am and the family members immediately sought care at the indigenous health unit closest to the village. From this unit, the boy was transferred to the indigenous health center at Belém do Solimões. This unit also does not have anti-ophidic serum and the boy was transferred to the nearest hospital, in Tabatinga, and was admitted at approximately 7 pm and received the *Bothrops* antivenom. The dead snake was taken to this health facility. Given the severity of the case, and evolution with extensive edema and compartment syndrome, the patient was taken to Manaus by plane. In Manaus, the condition worsened, and amputation of the left lower limb at the middle of the thigh was required. The boy stayed in Manaus for eight months and then returned to the village. The family consists of the parents and five children, and lives off fishing, hunting and subsistence farming. They produce cassava flour for their own consumption. 

The father reports that he has already been bitten by a snake and that his wife has been bitten twice. However, he did not go to the hospital and his wife only went to the hospital once when she was bitten, as she had edema and was in a lot of pain. He added that whenever possible they kill the snake that bit the person. He mentions that when transportation by canoes or motorized boats is available, the indigenous people of the village usually seek medical attention quickly at the nearest health unit. For prevention, the father emphasizes the need to always be attentive and careful during work activities. 

Currently, the boy attends Tikuna school regularly. The father carries the boy on his shoulders to take him to school. The father said the boy plays indoors with the other siblings. The father reports that the boy is interested in having a prosthesis to so he can walk again (Figure 3). 

### 2.3. Case 3

A 15-year-old Kubeo teenager, living in the peri-urban region of the municipality of São Gabriel da Cachoeira, Amazonas, was bitten by a jararaca when he was 5 years old. The interview conducted for this study was with the patient’s father, who is fluent in Portuguese. The boy was bitten on a trail as he returned from the fields with his grandmother, a nine-year-old brother, and other family members living in the village. According to the father, the boy liked to come running from the fields and ended up stepping on the snake right at the moment when it was crossing the trail. He adds that, at the time of the bite, many snakes were found on this trail: “Around the month of May, many snakes appeared due to the rains! So, every afternoon, when we returned from the fields, we found them and killed one or two, exactly at that time (around 5 in the afternoon)”. The father recalls that the boy’s grandmother told the boy’s brother that he had been bitten and suddenly fainted. “My mother takes controlled medicine. She freaked out. Then, she told me that she lost her balance, she was completely out of it when she saw her little grandson passed out. So, she asked his older brother to run to tell us (the parents). And, according to her, she only remembers this!” On the trail, just behind the boy, came a cousin, who was the one who noticed that the bite had occurred and killed the snake. The cousin carried the boy for a part of the route, approximately a kilometer. Another village boy who was with the group ran in front of the others to warn the boy’s parents in the village. They then ran from the village to find the group and to help carry the boy the rest of the way.

At home, the boy was treated with indigenous medicine. The father says that it was the eve of a holiday and, therefore, there were no cars on the road that passed near the village. The idea was to ask for help from a driver to take the boy to the hospital. “We waited until about nine o’clock at night, then gave up and went home. Indigenous medicine was used in the village “to prevent death”, according to the father. 

“The medicine eased the pain, but the bitten limb began to swell. The next day he had a swollen knee. His leg wasn’t moving anymore”.

Early in the morning the next day, which was a holiday (1 May; Labor Day in Brazil), the father saw a police car passing along the road, in the opposite direction from the city. He then stood by the side of the road to await the return of police officers to call for help. At approximately 9 am, they returned and took the boy to the hospital, together with the father. The leg was already very swollen, and tests in the hospital revealed kidney failure. He was given anti-*Bothrops* serum 15 h after the bite. According to the father, “The doctor said that his examination had shown renal failure, and that it was serious. The doctor gave him antivenom. For this reason, because of kidney failure, he was sent to Manaus”. However, he was only transported to Manaus by plane three days later. In Manaus, he was admitted to a pediatric hospital unit. Faced with the severity, he was taken to the intensive care unit. Fasciotomy for the treatment of compartment syndrome was performed. He remained in the hospital for three months for reconstructive surgeries and skin grafts. According to the father, “They did everything to save the leg, but the doctor said that the part affected by the venom was necrotized, even part of the bone, he explained”. After discharge from the hospital, he stayed in Manaus for another five months to receive care for the affected member, staying in a house of an evangelical church member. The same church provided medical and nursing care during this period. Return tickets to the village were also funded by the church. 

The boy suffered great tissue loss of the left lower limb and, as he grew, there were significant motor deficits and scoliosis due to posture deviation. In 2021, nine years after the bite, he had surgery for limb amputation and fitting of the mechanical prosthesis in the city of Campinas, in the state of São Paulo (Figure 4). The boy was asked if he would like to add something to the interview. He said his leg is growing and the prosthesis does not fit well anymore. The father said that calluses are forming where the prosthesis is attached to the limb. In the indigenous health unit, there are no professionals trained for this type of patient follow-up. The boy receives financial assistance in the form of social security payments.

## 3. Discussion

Disability is a complex and evolving concept, and involves aspects of body function and structure, as well as the individual’s ability to carry out basic activities without the benefit of assistance [30]. Some children with disabilities live in families and communities that attempt to eliminate social barriers resulting from disability, while others live in social contexts that do not strive to promote disabled children’s full participation in society [3,4,5]. The patients of our cases live in indigenous communities, where children enjoy fairly large mobility, which allows them to go to different houses and participate in activities in the forest and fields, and in almost all aspects of social life, including commemorations and rituals [31]. The loss of autonomy at such an early stage of life due to disability as a result of snakebites, as observed in this study, can deprive children of these sensory and social experiences and of learning their future roles in the village. Thus, without a family and community support network that continues to allow these children to continue experiencing social activities, these productive capacities will not be strengthened over time, thus preventing individual learning.

Because children accompany adults in fishing, hunting, extraction from plants, and farming activities, they are exposed as much as adults are to snakebites during these activities. In this study, it was observed that children were bitten during these activities and were helped by relatives. The species *Bothrops atrox*, the Amazonian lancehead, responsible for the cases presented here, is commonly found on trails in forests used by residents of the region in their extractivism activities [32]. According indigenous caregivers, a snakebite is an unpredictable event. In addition, the snake camouflages itself very well and it knows how to remain invisible to humans in the midst of the foliage. Therefore, snakebites can be prevented only if individuals are very attentive and careful during activities in the forest and kill the snake before the attack [33]. In this sense, children would be less attentive than adults, since there is an overlap between everyday activities and typical games at this age. In fact, the participants in this study put a lot of emphasis on finding the perpetrator of the bite in order to kill it. It seems that there is a belief among indigenous people that the death of snakes decreases the risk of further bites in areas frequented by people [33].

In common for the three cases of this study, the *Bothrops* snakebite evolved to compartment syndrome, secondary bacterial infection and extensive necrosis, leading to atrophy and amputation of the bitten limb. Local tissue damage is an important effect in human victims of *Bothrops* snakebites and, in the most severe cases, compartment syndrome appears as a dangerous complication due to the possibility of ischemia, tissue necrosis and neuropathy [34,35,36,37]. Pediatric cases of compartment syndrome from snakebites are reported worldwide [37,38,39,40], in clinical pictures very similar to those seen in this study. In the Brazilian Amazon, an analysis of cases reported by the epidemiological surveillance system revealed that children up to 12 years old had a significantly higher risk of severe local complications, such as compartment syndrome, which often result in amputations [41]. Although the outcomes after fasciotomy in children tend to be excellent [42,43,44], in our cases, the exceedingly large delay until the antivenom treatment was received and then for the fasciotomy to be performed prevented the preservation of the limb. Unfamiliar clinical scenarios of a compartment syndrome resulting from a snakebite complicated with a secondary bacterial infection, as seen in the three cases reported here, can lead to diagnostic delays of compartment syndrome in children [42,43]. In low- and middle-income countries, clinicians rely on a physical examination of the patient to diagnose compartment syndrome. Many non-invasive approaches such as ultrasound assessment have the potential to be used to diagnose compartment syndrome caused by snakebites [45,46].

Unfortunately, we found that poor access to rehabilitation services, which are generally only available in the state capital, lead to prolonged hospitalization of patients with severe snakebite, which distances them from their territory and family and community ties. In addition, upon return to the village, patients are deprived of adequate assistance for rehabilitation, since this level of care is not present in the indigenous health districts. It is noteworthy that one of the cases described in this study managed to obtain a prosthesis and other rehabilitation procedures after intense articulation of the research team with the indigenous health district, which led the boy to being referred to a public service in another state. As observed, with the growth of the child, there is a need to adapt the prosthesis from time to time, but there are no professionals qualified to perform such a procedure in indigenous health units [47]. In this extremely vulnerable context, family care is extremely important and includes, in the case of indigenous people, a diverse arsenal of indigenous medicine [33].

For indigenous peoples, hospital treatment is not understood as the final stage of treatment for snakebites. Actually, a re-establishment of a sense of well-being and reintroduction of the individual into social life is planned upon the patients return to the village. This phase aims at the organic, social and psychological rehabilitation of the individual, and includes rituals (such as tobacco smoking and ayahuasca, for example), massages and compresses on the affected limb with animal fats and plant resins, and the oral or topical use of bitter plants [33]. At least in our work, contrary to some sources that say that people with disabilities are the target of stigma and exclusion in indigenous communities [48], in this study, no signs of these negative reactions were reported in the narratives of the participants. On the contrary, in the family context observed here, very strong bonds were witnessed between children with disabilities and their guardians in a dynamic that motivates caregivers and care recipients to assume and continue the caregiving relationship [49].

As confirmed in our case series, therapeutic itineraries of snakebite patients in the Brazilian Amazon are very fragmented and are marked by several changes in means of transport along the route [16]. Access to antivenom required a considerable effort on the part of the parents of the children that participated in this study, since in the health units of indigenous areas this immunobiological is unavailable. The main obstacles that were identified were poor access to health care due to long distances and geographic barriers, and lack of timely transport to hospitals. Decentralization of antivenom from reference hospitals to community health centers in the Brazilian Amazon would be an effective strategy that would maximize access to treatment with antivenom [11,17].

One limitation of this study is the incompleteness of clinical and laboratorial information regarding the cases, since all the information presented comes from the narratives of those responsible for the patients. However, since we already understand much of the pathophysiology and factors associated with the severity of *B. atrox* envenomation [19], we believe that the most innovative part of our results is related to better understanding the social burden of the problem. These illness narratives can be used in medical education and may have implications for clinicians’ thinking and care of patients since they enhance clinicians’ sensitivity to the problem [50]. 

## 4. Conclusions

Snakebites are the cause of disabilities in indigenous children in the Brazilian Amazon, and also affect the full development of these children, as well as causing social, economic and psychological impacts on the family. These disabilities could be prevented if antivenom treatment were incorporated into the indigenous health system, which would facilitate timely access to effective treatment. The results demonstrate that prospective studies should be conducted in the Amazon region to estimate the real burden of disabilities from snakebites, so that public policies for the treatment and rehabilitation of patients can be formulated based on accurate information, through culturally tailored interventions.

## Figures and Tables

**Figure 1 toxins-15-00352-f001:**
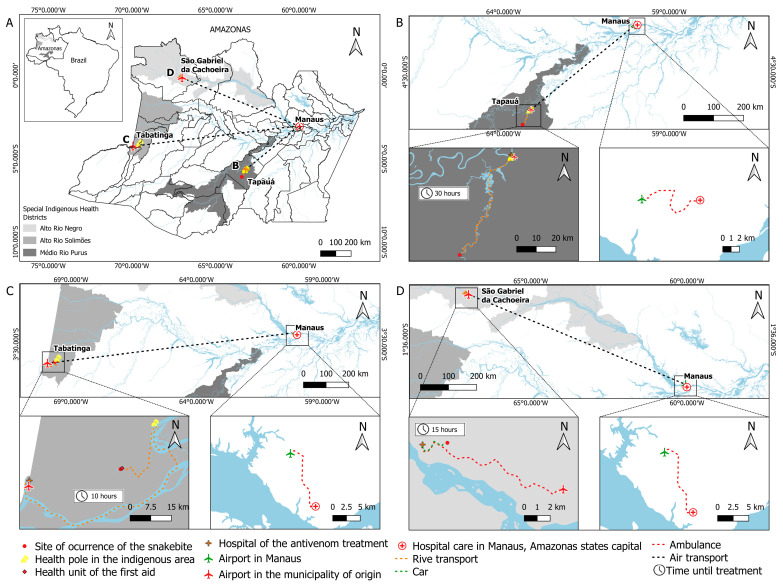
Therapeutic itineraries of the three children presenting severe physical disabilities from snakebite envenomations in indigenous villages of the Brazilian Amazon. On the maps, it is possible to follow the trajectory taken by each child who was a victim of a snakebite from the moment of the bite until they reach specialized care. (**A**) Amazonas state, in the northern region of Brazil; (**B**) Case 1: Apurinã boy took 30 h to receive antivenom treatment at the hospital in Tapauá; after antivenom treatment, the boy developed severe complications requiring transport to Manaus, capital of the Amazonas state, for surgery and specialized treatment. (**C**) Case 2: Boy of the Tikuna ethnic group took 10 h to receive the antivenom treatment at the hospital in Tabatinga, and then evolved with severe complications that require his transfer to Manaus for surgery and specialized treatment. (**D**) Case 3: Boy of the Kubeo ethnic group took 15 h to receive the antivenom treatment at the hospital of São Gabriel da Cachoeira, and then developed severe complications that required his transfer to Manaus.

**Figure 2 toxins-15-00352-f002:**
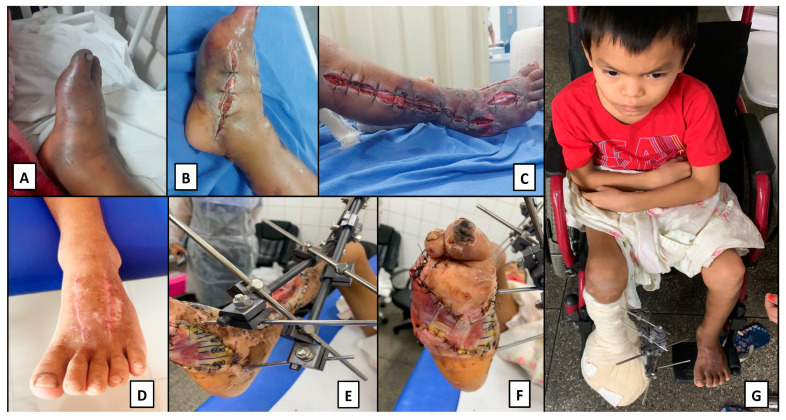
Photographs of the Apurinã boy, bitten by a jararaca in a village in the municipality of Tapauá, AM. He had access to antivenom; though with a delay of 30 h after the bite. (**A**)—Left lower limb with compartment syndrome at the time of arrival in Manaus, AM; (**B**)—left foot after emergency fasciotomy; (**C**)—left lower limb after emergency fasciotomy; (**D**)—right foot with scar 1 month after fasciotomy; (**E**)—left lower limb 4 months after fasciotomy and orthopedic external fixation; (**F**)—left foot in healing phase after 4 months of fasciotomy and amputation of toes; (**G**)—patient in a wheelchair.

**Figure 3 toxins-15-00352-f003:**
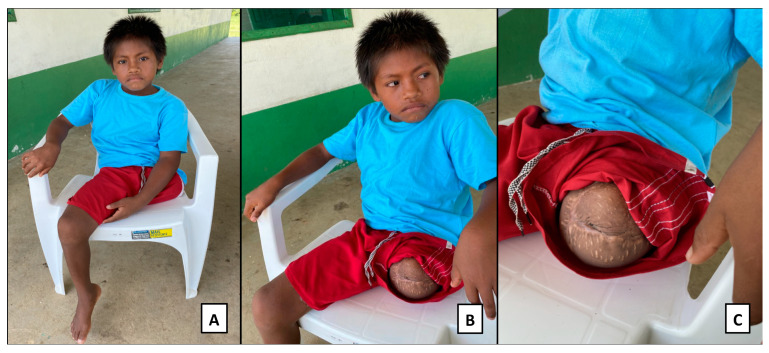
Photographs (**A**–**C**) show amputation in three different planes of the leg of an indigenous Tikuna boy bitten by a jararaca at the age of 7 in the village where he lives. After being transferred to the hospital in the urban area of Tabatinga (AM), he obtained access to antivenom after a delay of 19 h.

**Figure 4 toxins-15-00352-f004:**
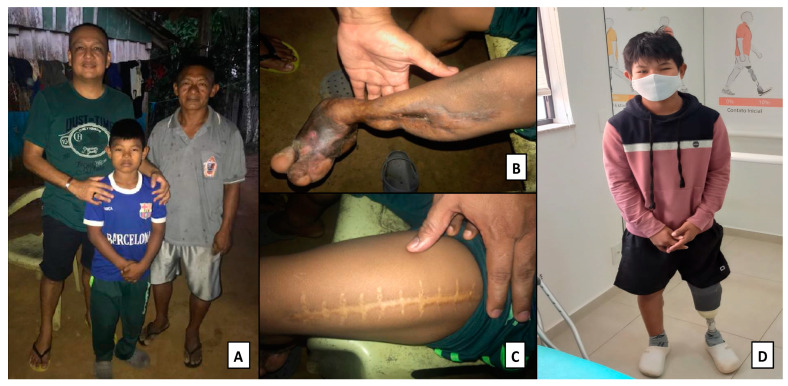
Indigenous Kubeo boy, resident of the municipality of São Gabriel da Cachoeira (AM), bitten by a jararaca at the age of 5 when returning from the fields with family members. (**A**)—Visit to the victim and his family members in 2019 for disability evaluation; (**B**)—atrophied left lower limb as a result of secondary infections; (**C**)—extensive scar on the right thigh due to the fasciotomy performed in a children’s hospital in the city of Manaus (AM); (**D**)—patient using prosthesis 9 years after the bite.

**Table 1 toxins-15-00352-t001:** Characteristics of the children with disabilities and their therapeutic itinerary.

Participant Information	Case 1	Case 2	Case 3
Gender	Male	Male	Male
Age	9 years	7 years	5 years
Municipality	Tapauá	Tabatinga	São Gabriel da Cachoeira cachoeira
Ethnicity	Apurinã	Tikuna	Kubeo
Perpetrating snake	*Bothrops atrox*	*Bothrops atrox*	*Bothrops atrox*
Place of occurrence	In the field collecting fruits with his family members	Playing in the village	On a trail, returning from farming activities with family members
Time from snakebite to arrival in the village	1 h	…	1 h
Time waiting for transport from the village to the indigenous health center	27 h	4.5 h	13 h
Time from the village to the indigenous health center	1 h	1.5 h	…
Time from the indigenous health center to the arriving at the hospital to receive antivenom	1 h	4 h	1 h
Time from snakebite to arriving at the hospital to receive antivenom	30 h	10 h	15 h
Clinical picture of the snakebite	Intense pain, extensive edema, compartment syndrome, secondary bacterial infection and extensive necrosis	Intense pain, extensive edema, compartment syndrome, secondary bacterial infection and extensive necrosis	Intense pain, extensive edema, compartment syndrome, secondary bacterial infection and extensive necrosis
Disabilities	Atrophy and loss of function of the right lower limb loss of left lower limb mobility	suprapatellar amputation of the left lower limb	Atrophy of the left lower limb with later suprapatellar amputation for fitting of a prosthesis
Use of prosthesis	…	No	Yes

## Data Availability

Not applicable.

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
