# Peer review of "Children Growing Up with Severe Disabilities as a Result of Snakebite Envenomations in Indigenous Villages of the Brazilian Amazon: Three Cases and Narratives"

_toxins, 2023, doi:10.3390/toxins15060352_

Round 1

Reviewer 1 Report

Dear Authors,

The present study presents three cases of children in indigenous villages of the Brazilian Amazon presenting severe physical disabilities from snakebite envenomations. The research subject is interesting and brings scientific important data in the field. Some changes of the manuscript should nevertheless be performed in order to improve its quality. Following specific changes should thus be performed:

 Major changes

Abstract: It should follow the structure of the manuscript, having the same structure.

Introduction: In this section, authors need to compare the purposes of the present study with similar studies existing in literature. Afterwards, you need to clarify what you bring in novelty. It is very important to state what exactly you bring in novelty in order to express your originality. The purpose of the study needs to be found in the last paragraph and be clearer. Please add further information and justifications and modify accordingly.

Discussions: This section is absolutely mandatory to be improved. It does not highlight the findings of your study and neither connections between the three presented cases are found. In fact, the part which needs to be the most consistent is not at all consistent. Moreover, novelty and originality of your study is essential to be emphasized in this section once again. You need to emphasize this in terms of results, not purposes, as the Introduction should. Please modify accordingly. In my opinion, if a more adequate presentation (especially in this part which needs more connections between the presented cases, from all points of view) of the study is not performed, the manuscript remains a case presentation, which is not necessarily interesting for the journal.

Conclusions: Please add perspectives of your study.

References should follow the recommendations of the journal and they are very few in number.

All these suggested changes should be performed in order to bring further improvements to the manuscript.

The English language is fine, only minor revisions should be performed.

Author Response

Dear Reviewer 1,

Thanks for the comments, which helped a lot in improving the manuscript. Firstly, we should point out that the manuscript was changed from the category 'Article' to 'Case report'. With this, we believe that many of this reviewer's recommendations have already been resolved, such as the reference number.

Abstract: It should follow the structure of the manuscript, having the same structure.

RESPONSE: Done.

Introduction: In this section, authors need to compare the purposes of the present study with similar studies existing in literature. Afterwards, you need to clarify what you bring in novelty. It is very important to state what exactly you bring in novelty in order to express your originality. The purpose of the study needs to be found in the last paragraph and be clearer. Please add further information and justifications and modify accordingly.

RESPONSE: Done.

Discussions: This section is absolutely mandatory to be improved. It does not highlight the findings of your study and neither connections between the three presented cases are found. In fact, the part which needs to be the most consistent is not at all consistent. Moreover, novelty and originality of your study is essential to be emphasized in this section once again. You need to emphasize this in terms of results, not purposes, as the Introduction should. Please modify accordingly. In my opinion, if a more adequate presentation (especially in this part which needs more connections between the presented cases, from all points of view) of the study is not performed, the manuscript remains a case presentation, which is not necessarily interesting for the journal.

RESPONSE: The new version of the manuscript has 6 paragraphs, including one about the limitations of the work, which we consider a considerably long text for a 'Case report' type manuscript. We address more in the Discussion the social aspects of the results, since the information was collected through interviews and presented in forms of narratives of the guardians or the children.

The following topics are discussed: Disability as a social barrier, mobility of children in indigenous communities, possibility of loss of autonomy at such an early stage of life, family and community support network, children exposure to snakebites in the indigenous villages, snakebite circumstances in indigenous children, poor access to rehabilitation services, indigenous medicine, caregiving relationship, therapeutic itineraries of snakebite patients in the Brazilian Amazon, decentralization of antivenom from reference hospitals to community health centers as a potential strategy to improve access and reduce snakebites burden, limitations of the study, use of medical narratives.

Conclusions: Please add perspectives of your study.

RESPONSE: Done.

References should follow the recommendations of the journal and they are very few in number.

RESPONSE: See above.

Sincerely,

Reviewer 2 Report

The work presented addresses relevant aspects, considering the problem that ophidian accidents represent in the Amazonian population. In addition, it presents a proper narration and the reading is interesting.

Considering the above, it is suggested that the manuscript be accepted for publication.

Author Response

Dear reviewer 2,

We very much appreciate comments on the manuscript.

Sincerely,

Reviewer 3 Report

I read the paper entitled " Children Growing Up with Severe Disabilities as a Result of Snakebite Envenomations in Indigenous Villages of the Brazilian Amazon:  Three Cases and Narratives" and submitted for publication in Toxins.
The paper describes three cases of child snake bites between indigenous villages of the Brazilian Amazon which led to severe motor disability. This involves in addition to the problems of a medical and rehabilitative nature also problems of a social nature related to the difficulty of the disabled in participating in an active social life (and school) in the absence of a correct, accessible and above all continuous medical-rehabilitative support. These issues, together with the role of disabled children in the social context of the villages in which they live, are discussed in the paper.

I found the paper interesting and articulated above all in framing the social and medical-social problems related to the difficulties and therefore to the delayed treatment of snake bites in the Amazon region.
In my opinion, there is a lack of a clearer vision of the problem for the reader through tables that recall numbers and/or maps that recall the geographical dimension of the areas where the cases described occurred and the places where medical treatment is accessible.

For this reason, I suggest accepting the publication of the paper in this journal after minor revisions

Author Response

Dear Reviewer 3,

Thank you so much for your comments.

As suggested, we added a new Figure (map with therapeutic itineraries of the 3 cases) and a Table (summarizing data from the 3 cases) to this new version of the manuscript.

Sincerely,

Round 2

Reviewer 1 Report

Dear Authors,

The present study presents three cases of children in indigenous villages of the Brazilian Amazon presenting severe physical disabilities from snakebite envenomations. The authors performed most of the suggested changes after the first round of review. However, following specific changes should still be performed:

Major changes

Abstract: A conclusion/perspectives phrase should be added.

Introduction: I still cannot find the part where you describe the novelty of your study and regarding the purpose of the study, I still did not find the secondary purpose of the study which concerns the comparison of the three cases.

Discussions: Novelty and originality of your study is not emphasized in this section once again. You need to emphasize this in terms of results, not purposes, as the Introduction should.

References should follow the recommendations of the journal and they are very few in number.

All these suggested changes should be performed in order to bring further improvements to the manuscript. 

English language is fine, it just needs minor corrections.

Author Response

It is a pleasure for us to respond for the second time to the questions raised about our manuscript, in order to make it better.

Abstract: A conclusion/perspectives phrase should be added.

RESPONSE: Done accordingly.

Introduction: I still cannot find the part where you describe the novelty of your study and regarding the purpose of the study, I still did not find the secondary purpose of the study which concerns the comparison of the three cases. 

RESPONSE:

These sentences are added in the last paragraph of the Introduction.

"Similarly, recent studies conducted in Brazil demonstrate a much higher rate of snakebite envenomations in indigenous populations of the Brazilian Amazon [11,29]. This ecological association raises the urgent need to investigate the burden of disabilities caused by snakebites among indigenous populations. In general, previous studies on disabilities from snakebites carried out in the Amazon described only the clinical aspects of the problem (19,25-27), without stressing the repercussions of the problem on the social life of patients as intended in this study."

Discussions: Novelty and originality of your study is not emphasized in this section once again. You need to emphasize this in terms of results, not purposes, as the Introduction should.

RESPONSE: We added one paragraph to the Discussion to explain about the clinical aspects of the observed complications, especially the compartment syndrome. This is a complication observed for the 3 cases, not valued in the previous version of the manuscript. In addition, we try to highlight the findings of the study in this same section.

References should follow the recommendations of the journal and they are very few in number.

RESPONSE: Done accordingly.

English language is fine, it just needs minor corrections.

RESPONSE: Done accordingly.

Best wishes,

Wuelton Monteiro